

# Characterization of endoplasmic reticulum-associated degradation in the human fungal pathogen *Candida albicans*

Ellen M. Doss[1,2], Joshua M. Moore[1], Bryce H. Harman[1],
Emma H. Doud[3,4], Eric M. Rubenstein[1] and Douglas A. Bernstein[1]

[1] Department of Biology, Ball State University, Muncie, Indiana, United States
[2] Mode of Action and Resistance Management Center of Expertise, Corteva Agriscience, Indianapolis, Indiana, United States
[3] Center for Proteome Analysis, Indiana University School of Medicine, Indianapolis, Indiana, United States
[4] Department of Biochemistry and Molecular Biology, Indiana University School of Medicine, Indianapolis, Indiana, United States

Corresponding authors
Eric M. Rubenstein,
emrubenstein@bsu.edu
Douglas A. Bernstein,
dabernstein@bsu.edu

## ABSTRACT

**Background:** *Candida albicans* is the most prevalent human fungal pathogen. In immunocompromised individuals, *C. albicans* can cause serious systemic disease, and patients infected with drug-resistant isolates have few treatment options. The ubiquitin-proteasome system has not been thoroughly characterized in *C. albicans*. Research from other organisms has shown ubiquitination is important for protein quality control and regulated protein degradation at the endoplasmic reticulum (ER) *via* ER-associated protein degradation (ERAD).

**Methods:** Here we perform the first characterization, to our knowledge, of ERAD in a human fungal pathogen. We generated functional knockouts of *C. albicans* genes encoding three proteins predicted to play roles in ERAD, the ubiquitin ligases Hrd1 and Doa10 and the ubiquitin-conjugating enzyme Ubc7. We assessed the fitness of each mutant in the presence of proteotoxic stress, and we used quantitative tandem mass tag mass spectrometry to characterize proteomic alterations in yeast lacking each gene.

**Results:** Consistent with a role in protein quality control, yeast lacking proteins thought to contribute to ERAD displayed hypersensitivity to proteotoxic stress. Furthermore, each mutant displayed distinct proteomic profiles, revealing potential physiological ERAD substrates, co-factors, and compensatory stress response factors. Among candidate ERAD substrates are enzymes contributing to ergosterol synthesis, a known therapeutic vulnerability of *C. albicans*. Together, our results provide the first description of ERAD function in *C. albicans*, and, to our knowledge, any pathogenic fungus.

## INTRODUCTION

*Candida albicans* is a commensal fungus residing in the gastrointestinal tract of humans (*Kumamoto, Gresnigt & Hube, 2020*). *C. albicans* is found in or on most humans and can cause superficial infections, such as oral thrush or vaginosis (*Goncalves et al., 2016*; *Millsop & Fazel, 2016*). However, in immunocompromised individuals, *C. albicans* can cause more serious infections such as candidemia and candidiasis, infections of blood or hard organs, respectively (*Pappas et al., 2018*). These infections can be life-threatening when left untreated. There are a number of antifungal drugs that can be used to treat *C. albicans* infection; however, drug-resistant isolates have been and continue to be isolated, limiting treatment options (*Lee et al., 2021*). As such, it is critical that we develop novel antifungal therapeutics to treat these resistant infections. To do this, we must better understand *C. albicans* function at the molecular level.

One area of *C. albicans* biology that has yet to be thoroughly characterized is the ubiquitin-proteasome system (UPS). The UPS plays numerous important roles in eukaryotic cells, degrading proteins that are either no longer needed or have been compromised (*e.g.*, by misfolding) (*Berner, Reutter & Wolf, 2018*). In *C. albicans*, ubiquitin is important for stress adaptation, and deletion of the gene that encodes ubiquitin, *Ca_UBI4*, attenuates virulence in a mouse model (*Leach et al., 2011*). Furthermore, mutation of *Ca_UBI4* leads to hypersensitivity to oxidative stress, the antifungal drug caspofungin, and the ER stress inducer tunicamycin. Ubiquitylation plays important roles in protein quality control and regulated protein degradation at the endoplasmic reticulum (ER) *via* ER-associated protein degradation (ERAD) (*Berner, Reutter & Wolf, 2018*; *Mehrtash & Hochstrasser, 2019*); as such, ERAD may be a vulnerability that can be leveraged during antifungal therapeutic discovery.

ERAD has been extensively characterized in the model unicellular fungal eukaryote, *Saccharomyces cerevisiae*, and in mammals. The major ERAD ubiquitin ligases in *S. cerevisiae* are the multipass transmembrane proteins ScHrd1 and ScDoa10 (Fig. 1). ScHrd1 primarily functions with the ubiquitin-conjugating enzyme ScUbc7 (which is anchored at the ER membrane by the transmembrane protein ScCue1), while ScDoa10 works with two ubiquitin-conjugating enzymes, ScUbc7 and the transmembrane enzyme ScUbc6 (*Bays et al., 2001*; *Lips et al., 2020*; *Plemper et al., 1999*; *Swanson, Locher & Hochstrasser, 2001*). ScHrd1 and ScDoa10 target distinct, but partially overlapping, subsets of protein quality control substrates for degradation. ScHrd1 mediates turnover of ER soluble luminal proteins, transmembrane proteins, and proteins that persistently engage (*i.e.*, clog) the ER translocon, while ScDoa10 promotes destruction of soluble cytosolic and transmembrane proteins (*Carvalho, Goder & Rapoport, 2006*; *Gauss, Sommer & Jarosch, 2006*; *Habeck et al., 2015*; *Huyer et al., 2004*; *Metzger et al., 2008*; *Ravid, Kreft & Hochstrasser, 2006*; *Rubenstein et al., 2012*; *Runnebohm et al., 2020b*; *Sato et al., 2009*). In mammals, ERAD is mediated by an expanded cadre of ubiquitin ligases and conjugating enzymes, including homologs of ScHrd1 (HRD1/SYVN1 and gp78) and ScDoa10 (MARCHF6/TEB4) (*Hassink et al., 2005*; *Kikkert et al., 2004*; *Liang et al., 2003*). In addition to their well-characterized roles in protein quality control, ScHrd1 and
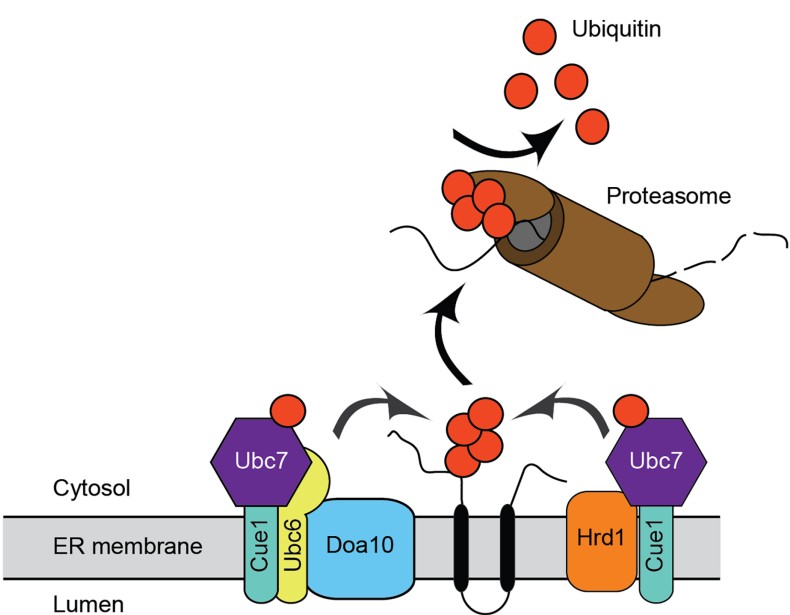

**Figure 1** **Endoplasmic reticulum-associated degradation.** Mechanism for ER protein degradation in *S. cerevisiae*. See text for details.

ScDoa10 also contribute to regulated turnover of otherwise normal proteins. For example, *S. cerevisiae* and mammalian homologs of Hrd1 and Doa10 promote regulated turnover of sterol biosynthetic enzymes (*Foresti et al., 2013*; *Garza, Tran & Hampton, 2009*; *Hampton, Gardner & Rine, 1996*; *Huang & Chen, 2023*; *Jo et al., 2011*; *Stevenson, Huang & Olzmann, 2016*).

Virtually nothing is known about ERAD mechanisms in pathogenic fungi, including *C. albicans*. *Candida glabrata UBC7* mRNA abundance exhibits a ~3-fold increase following azole treatment, suggesting a role for ERAD in pathogenic fungal stress response (*Li, Skinner & Bennett, 2012*). In this study, we performed initial characterization of *Ca_HRD1*, *Ca_DOA10*, and *Ca_UBC7*, genes predicted to encode enzymes that function in ERAD. We generated homozygous *C. albicans* mutants lacking *Ca_HRD1*, *Ca_DOA10*, and *Ca_UBC7*. Consistent with a role in protein quality control, yeast lacking putative ERAD components exhibited hypersensitivity to proteotoxic stress. Further, mass spectrometric analyses of wild type and mutant yeast revealed distinct proteomic profiles for each mutant, illuminating candidate physiological ERAD substrates, co-factors, and compensatory stress response factors. Together, these results provide the first description of ERAD function in *C. albicans*, and, to our knowledge, any pathogenic fungus.

## MATERIALS AND METHODS

### Generation of mutants

Mutations to *Ca_HRD1*, *Ca_DOA10*, and *Ca_UBC7* were made using CRISPR-mediated genome editing. Guide RNAs to the 5′ region of *Ca_HRD1*, *Ca_DOA10*, and *Ca_UBC7* were cloned into plasmid pV1093 (*Vyas, Barrasa & Fink, 2015*), which also contains Cas9 as well as nourseothricin (Nat) resistance marker (*Nat^R*). Repair templates that introduce a

stop codon and restriction site upon homologous recombination with the cut chromosome were generated. *C. albicans* wild type strain SC5314 was transformed with Cas9 guide constructs and repair templates using lithium acetate. Transformants were selected on 200 μg/ml Nat. Correct homozygous transformants were identified by colony PCR followed by restriction digestion. Sanger sequencing was used to confirm the mutants. Homozygous mutants will be henceforth referred to as *Ca_hrd1*, *Ca_doa10*, and *Ca_ubc7*.

## Growth analyses

**Spot assays:** Yeast growth assays were performed using a modified version of a previously published protocol (*Watts et al., 2015*). Yeast were grown to saturation overnight at 30 °C in yeast extract-peptone-dextrose media (YPD). Cells were diluted to an $OD_{600}$ of 1.0. Fivefold dilutions were prepared in a 96-well plate in YPD. Cells were plated using a pin replicator onto YPD agar lacking or containing increasing concentrations of hygromycin B (*Brodersen et al., 2000*; *Ganoza & Kiel, 2001*). Cells were grown at 30 and 37 °C for 5 days and at 25 and 40 °C for 10 days.

**Inhibition assays:** A modified drug susceptibility assay (*Bauer et al., 1966*) was performed. Yeast were grown to saturation overnight at 30 °C in YPD. $1 \times 10^7$ cells were spread on 20 ml of synthetic complete media in 150 mM plates and were allowed to dry. A sterile filter paper disc was placed on the cells, and 15 μl of 1 M dithiothreitol (DTT) suspended in water, 33% (v/v) β-mercaptoethanol (βME) in water (*Jia et al., 2019*), or sterile water were directly pipetted onto the paper disc. Plates were incubated for 3 days at 37 °C before visualization.

## Mass spectrometry and data analysis

Sample preparation, mass spectrometry analysis, bioinformatics, and data evaluation for quantitative proteomics experiments were performed in collaboration with the Indiana University Proteomics Center for Proteome Analysis at the Indiana University School of Medicine (IUSM) similarly to previously published protocols (*Kumar et al., 2022*; *Morris et al., 2022*; *Soundararajan et al., 2022*; *Stanhope et al., 2023*).

**Sample preparation:** 12 samples (*n* = 3 of wild type, *Ca_hrd1/Ca_hrd1*, *Ca_doa10/Ca_doa10*, and *Ca_ubc7/Ca_ubc7* yeast) were submitted to the IUSM Center for proteome analysis, where proteins were denatured in 8 M urea, 100 mM Tris-HCl, pH 8.5 with sonication using a Bioruptor® sonication system (Diagenode Inc, Denville, NJ, USA) with 30 s/30 s on/off cycles for 15 min in a water bath at 4 °C. After subsequent centrifugation at 14,000 rcf for 20 min, protein concentrations were determined by Bradford protein assay (BioRad Cat No: 5000006). A total of 100 μg equivalent of protein from each sample were reduced with 5 mM tris (2-carboxyethyl) phosphine hydrochloride (TCEP, Sigma-Aldrich Cat No: C4706) for 30 min at room temperature and alkylated with 10 mM chloroacetamide (CAA, Sigma Aldrich Cat No: C0267) for 30 min at room temperature in the dark. Samples were diluted with 50 mM Tris.HCl, pH 8.5 to a final urea concentration of 2 M for Trypsin/Lys-C based overnight protein digestion at 37 °C (1:70 protease: substrate ratio, Mass Spectrometry grade, Promega Corporation, Cat No: V5072.).

**Peptide purification and labeling:** Digestions were acidified with trifluoroacetic acid (TFA, 0.5% v/v) and desalted on Sep-Pak® Vac cartridges (Waters™ Cat No: WAT054955) with a wash of 1 ml 0.1% TFA followed by elution in 70% acetonitrile 0.1% formic acid (FA). Peptide concentrations were checked by Pierce Quantitative colorimetric assay (Cat No: 23275) and confirmed to be consistent across all samples. 50 μg peptides were then labeled with 0.25 mg Tandem Mass Tag pro (TMTpro) reagent (Thermo Fisher Scientific, TMTpro™ Isobaric Label Reagent Set; Cat No: A44520, Lot VL313890; see Table S1) for 2 h at room temperature, quenched with a final concentration v/v of 0.3% hydroxylamine at room temperature for 15 min. Labeled peptides were mixed and dried by speed vacuum.

**High pH basic fractionation**: For high pH basic fractionation, peptides were reconstituted in 0.1% trifluoroacetic acid and half of the mixture was fractionated on a 50 mg Sep-Pak® Vac cartridge using methodology and reagents from Pierce™ High pH reversed-phase peptide fractionation kit (Thermo Fisher Cat No: 84868).

**Nano-LC-MS/MS**: Global proteomics were performed on an EASY-nLC 1200 HPLC system (SCR: 014993; Thermo Fisher Scientific, Waltham, MA, USA) coupled to Lumos Orbitrap™ mass spectrometer (Thermo Fisher Scientific, Waltham, MA, USA). A total of 1/5 of each fraction was loaded onto a 25-cm EasySpray column (Thermo Fisher Scientific ES902A, Waltham, MA, USA) at 400 nl/min. Peptides were eluted from 4–30% with mobile phase B (Mobile phases A: 0.1% FA, water; B: 0.1% FA, 80% Acetonitrile (Thermo Fisher Scientific Cat No: LS122500, Waltham, MA, USA)) over 160 min, 30–80% B over 10 min, and dropping from 80–10% B over the final 10 min. The mass spectrometer was operated in positive ion mode with a 4-s cycle time data-dependent acquisition method with advanced peak determination and Easy-IC (internal calibrant) on. Precursor scans (m/z 375–1,600) were done with an orbitrap resolution of 120,000, RF lens% 30, maximum inject time 50 ms, AGC target of 100% (4e5), MS2 intensity threshold of 2.5e4, MIPS mode to peptide, including charges of 2 to 7 for fragmentation with 30 s dynamic exclusion. MS2 scans were performed with a quadrupole isolation window of 0.7 m/z, 37% HCD CE, 50,000 resolution, 200% normalized AGC target, dynamic maximum IT, fixed first mass of 100 m/z.

**Mass spectrometry data analysis**: Resulting RAW files were analyzed in Proteome Discover™ 2.5 (Thermo Fisher Scientific, Waltham, MA, USA) with a *C. albicans* UniProt FASTA (downloaded 03/15/2022, 6,030 entries) plus common contaminants. SEQUEST HT searches were conducted with a maximum number of three missed cleavages, precursor mass tolerance of 10 ppm, and a fragment mass tolerance of 0.02 Da. Static modifications used for the search were carbamidomethylation on cysteine residues and TMTpro label on lysine residues. Dynamic modifications included oxidation of methionine, TMTpro label on peptide N terminus, and acetylation, methionine-loss, or methionine-loss plus acetylation on protein N terminus. Percolator False Discovery Rate (FDR) was set to a strict peptide spectral match FDR setting of 0.01 and a relaxed setting of 0.05. In the consensus workflow, peptides were normalized by total peptide amount with no scaling. Co-isolation thresholds of 50% and average reporter ion S/N cutoffs of five were used for quantification. All peptides were used for normalization and protein roll-up and

modified peptides were excluded in the pairwise ratio calculation. Protein abundance-based ratio calculations were done with no imputation. Protein FDR validator node was set to a strict target FDR of 0.01 and relaxed of 0.05. Resulting normalized abundance values for each sample type, abundance ratio and log2 (abundance ratio) values, and respective p-values (protein abundance-based ratio calculation and individual protein ANOVA) from Proteome Discover™ were exported to Microsoft Excel.

## RESULTS

### Structural analyses of ERAD enzymes

We identified putative *C. albicans* genes encoding ERAD enzymes CaHrd1, CaDoa10, and CaUbc7. A summary of amino acid sequence identity and similarity between homologous proteins in *C. albicans*, *S. cerevisiae*, and *H. sapiens* is presented in Table S2. Domain organization and AlphaFold-predicted (*Jumper et al., 2021*) or experimentally determined (*Arai et al., 2006*; *Cook et al., 1997*; *Wu et al., 2020*) structures of these enzymes are presented in Fig. 2.

All three (*H. sapiens*, *S. cerevisiae*, and *C. albicans*) Hrd1 homologs possess an N-terminal transmembrane domain with eight predicted membrane-spanning segments, a catalytic Really Interesting New Gene (RING) domain, and a largely unstructured C-terminal extension (CTE), which likely contributes to substrate and cofactor interactions (Figs. 2A and 2B) (*Omura et al., 2006*; *Schulz et al., 2017*). The predicted CaHrd1 CTE possesses *H. sapiens*-like and *S. cerevisiae*-like features. The CaHrd1 CTE includes an alpha helix with amphipathic character (green helix in Fig. 2B) resembling the human C-terminal HAF-H domain implicated in HsHRD1 complex formation (*Schulz et al., 2017*). Like the *S. cerevisiae* enzyme, the CaHrd1 CTE includes a predicted two-stranded beta-sheet (cyan strands in Fig. 2B) with uncharacterized function.

AlphaFold-predicted structures of *H. sapiens*, *S. cerevisiae*, and *C. albicans* Doa10/MARCHF6 proteins include N-terminal RING domains, large C-terminal transmembrane domains, and short N- and C-terminal extensions (Figs. 2A and 2C). The transmembrane portions of these proteins include a conserved three-transmembrane segment TEB4-Doa10 (TD) domain (cyan in Fig. 2C) (*Kreft & Hochstrasser, 2011*). The large transmembrane portion of Doa10 homologs has been proposed to function as a retrotranslocation channel for ER export of transmembrane substrates, with the first (yellow) and fourteenth (green) transmembrane segments forming a lateral gate for substrate entry (*Mehrtash & Hochstrasser, 2022*; *Schmidt, Vasic & Stein, 2020*). The ScDoa10 and HsMARCHF6 CTEs promote substrate ubiquitylation (*Mehrtash & Hochstrasser, 2022*; *Zattas et al., 2016*); AlphaFold-guided mutational analyses indicate interactions between the CTE and N-terminal RING domain are essential for optimal ScDoa10 function (*Mehrtash & Hochstrasser, 2022*). AlphaFold structural predictions indicate this interaction is likely also present in CaDoa10 and HsMARCHF6. Alignments of Hrd1 and Doa10 homolog RING domains demonstrate conservation of zinc-coordinating cysteine and histidine residues as well as a tryptophan residue commonly found in ubiquitin ligase catalytic domains (Figs. 2B and 2C). Complete alignments of Hrd1 and Doa10 homologs are presented in Supplemental Files 1 and 2.

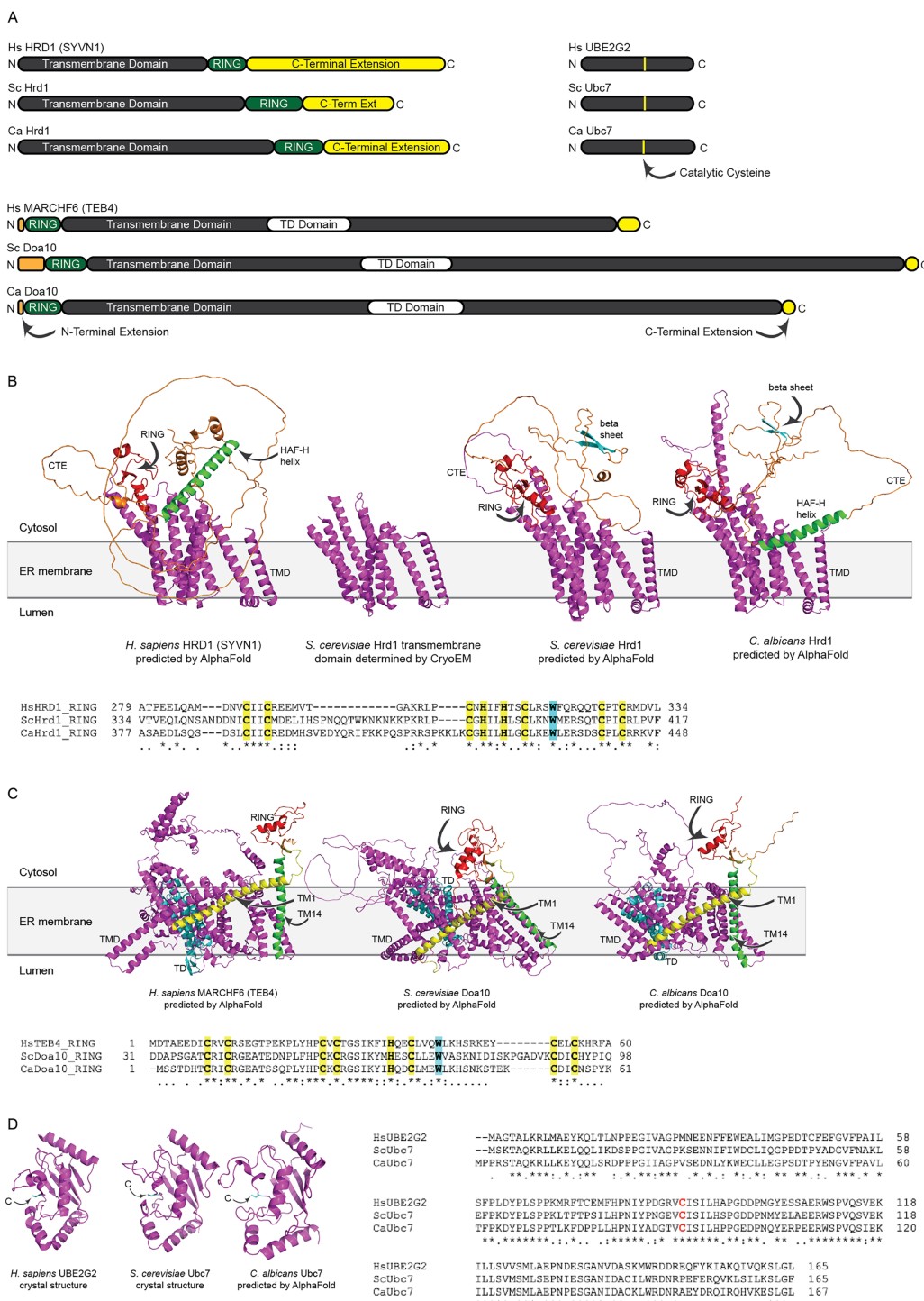

**Figure 2 Structural analysis of *H. sapiens*, *S. cerevisiae*, and *C. albicans* Hrd1, Doa10, and Ubc7 homologs.** (A) To-scale domain architecture of human and fungal homologs of Hrd1, Doa10, and Ubc7. See text for details. RING, Really Interesting New Gene domain. TD, TEB4-Doa10 domain. (B) Top, AlphaFold-predicted structures of *H. sapiens* HRD1, *S. cerevisiae* Hrd1, and *C. albicans* Hrd1 and cryo-electron microscopy structure of transmembrane portion of *S. cerevisiae* Hrd1 (PDB 6VJZ) (*Wu et al., 2020*). Magenta, transmembrane domain (TMD). Red, RING domain. Orange, C-terminal extension (CTE). Green, predicted HAF-H helix shared by HsHRD1 and CaHrd1. Cyan, predicted beta sheet shared by ScHrd1 and CaHrd1. Bottom, Amino acid alignment of catalytic RING domains of Hrd1

**Figure 2** (continued)
homologs. Bold yellow, Zinc-coordinating Cys and His residues. Bold cyan, Trp residue commonly found in RING domains. (C) Top, AlphaFold-predicted structures of *H. sapiens* MARCHF6, *S. cerevisiae* Doa10, and *C. albicans* Doa10. Magenta, transmembrane domain (TMD). Cyan, TD domain. Red, RING domain. Orange, C-terminal extension. Yellow and Green, first and fourteenth transmembrane segments, respectively (TM1 and TM14). Bottom, Amino acid alignment of catalytic RING domains of Doa10 homologs. Bold yellow, Zinc-coordinating Cys and His residues. Bold cyan, Trp residue commonly found in RING domains. (D) Left, Crystal structures of *H. sapiens* UBE2G2 (*Arai et al., 2006*) and *S. cerevisiae* Ubc7 (PDB 2UCZ) (*Cook et al., 1997*) and AlphaFold-predicted structure of *C. albicans* Ubc7. Cyan, catalytic cysteine (C). Right, Amino acid alignment of Ubc7 homologs. Bold red, catalytic cysteine.

*H. sapiens*, *S. cerevisiae*, and *C. albicans* Ubc7 homologs exhibit strong conservation of primary and predicted tertiary structure (Figs. 2A and 2D). Comparison of crystal structures of ScUbc7 and HsUBE2G2 with AlphaFold-predicted CaUbc7 reveals all three adopt a characteristic ubiquitin-conjugating enzyme fold; the position of the invariant catalytic cysteine is indicated.

## Generation of mutants

We generated functional deletions of *Ca_HRD1*, *Ca_DOA10*, or *Ca_UBC7* using CRISPR-mediated genome editing. Stop codons were inserted at the 5′ ends of each of these genes, effectively creating knockouts of *Ca_HRD1*, *Ca_DOA10*, and *Ca_UBC7*. PCR and Sanger sequencing were used to confirm the correct mutations were made and that mutations were homozygous, preventing translation of both alleles' transcripts. Heterozygous *Ca_HRD1/Ca_hrd1* mutants were also isolated. Introduction of these mutations did not impact growth on YPD (Fig. 3).

## *C. albicans* lacking *HRD1*, *DOA10*, or *UBC7* exhibit sensitivity to proteotoxic stress

A quality control function for *C. albicans* genes encoding ERAD factors has not previously been reported. *S. cerevisiae* with mutations in protein quality control genes (including those encoding ERAD enzymes) exhibit elevated sensitivity to hygromycin B (*Crowder et al., 2015*; *Runnebohm et al., 2020a*; *Verma et al., 2013*; *Woodruff et al., 2021*). Hygromycin B distorts the ribosomal A site, thereby reducing translational fidelity, which is predicted to increase the abundance of aberrant proteins (*Brodersen et al., 2000*; *Ganoza & Kiel, 2001*). We compared the growth of wild type yeast and yeast with homozygous disruptions of *Ca_HRD1*, *Ca_DOA10*, or *Ca_UBC7* in the presence of increasing concentrations of hygromycin B across a range of temperatures (Fig. 3). Homozygous deletion of *Ca_HRD1*, *Ca_DOA10*, or *Ca_UBC7* caused a pronounced growth defect in the presence of hygromycin B. Sensitivity to hygromycin B was markedly enhanced at elevated temperatures.

We also compared hygromycin B sensitivity of yeast possessing homozygous and heterozygous mutations in *Ca_HRD1* (Fig. 4). Heterozygous *Ca_HRD1/Ca_hrd1* yeast exhibited wild type resistance to hygromycin B at 25, 30, and 37 °C, indicating a single copy of *Ca_HRD1* is sufficient to confer protection from hygromycin B at these

## hygromycin B (µg/ml)

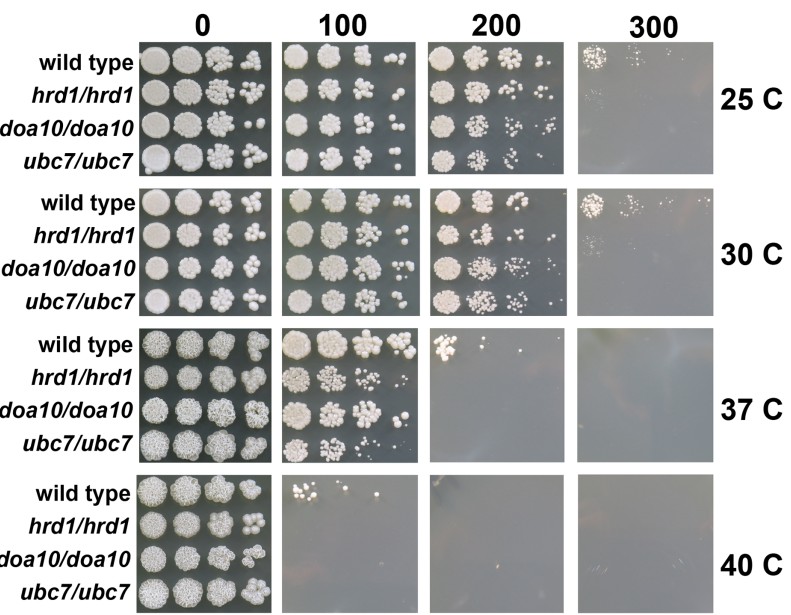

**Figure 3** *C. albicans* **ERAD enzymes confer resistance to hygromycin B.** Wild type or indicated homozygous mutant *C. albicans* were serially diluted and spotted onto YPD growth medium in the presence of increasing concentrations of hygromycin B and incubated at the indicated temperatures.

## hygromycin B (µg/ml)

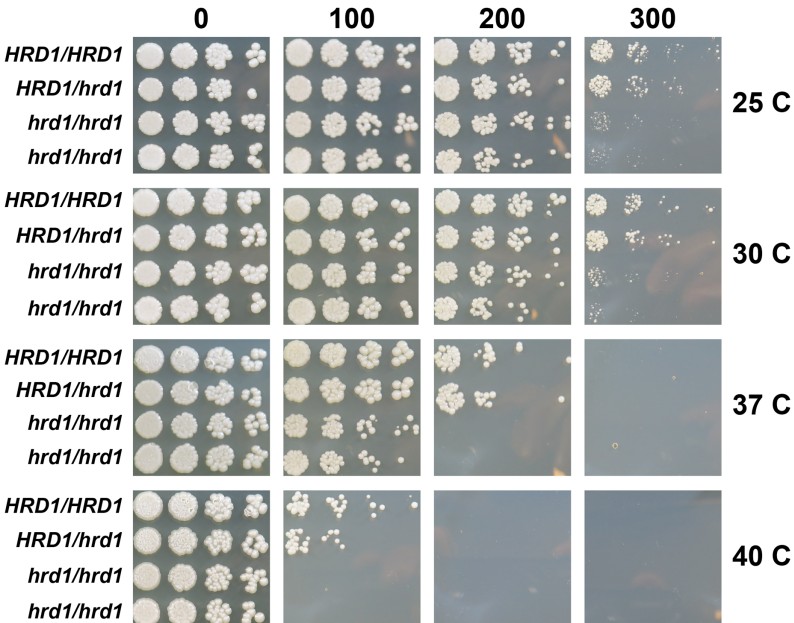

**Figure 4 A single copy of *C. albicans HRD1* is sufficient to confer resistance to hygromycin B.** Yeast of the indicated genotypes were serially diluted and spotted onto YPD growth medium in the presence of increasing concentrations of hygromycin B and incubated at the indicated temperatures.

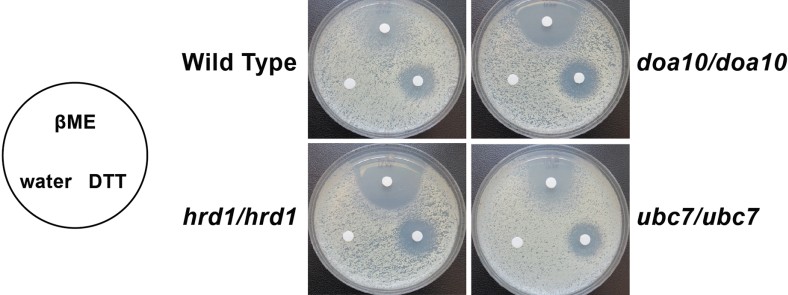

**Figure 5 *C. albicans* ERAD enzymes confer resistance to β-mercaptoethanol.** Lawns of yeast of the indicated genotypes were seeded onto YPD media. Prior to incubation at the indicated temperatures, sterile filter papers were impregnated with 15 μl sterile water, 33% β-mercaptoethanol (βME), or 1 M dithiothreitol (DTT).

temperatures. We observed a subtle growth defect of *Ca_HRD1/Ca_hrd1* yeast in the presence of hygromycin B at 40 °C.

We analyzed susceptibility of wild type yeast and ERAD mutants to drugs that cause ER stress (disulfide-reducing agents beta-mercaptoethanol (βME) and dithiothreitol (DTT)) (*Jia et al., 2019*; *Wu, Ng & Thibault, 2014*). Homozygous mutation of *Ca_HRD1*, *Ca_DOA10*, or *Ca_UBC7* enhanced sensitivity to βME (Fig. 5). By contrast, wild type and ERAD mutant strains exhibited similar susceptibility to DTT. Together, our results suggest *Ca_HRD1*, *Ca_DOA10*, or *Ca_UBC7* are important for *C. albicans* growth under proteotoxic stress.

## Proteomic analysis of ERAD mutant yeast strains

To identify candidate physiological substrates and interactors of ERAD enzymes, we identified proteins with altered abundance in yeast lacking *Ca_HRD1*, *Ca_DOA10*, or *Ca_UBC7* relative to wild type yeast using quantitative tandem mass tag (TMT) mass spectrometry. The complete results of these experiments are presented in Supplemental File 3 and summarized in Fig. 6, Tables 1–3 and Tables S2–S8. In total, 229 proteins exhibited significant changes in abundance in at least one of the three mutants analyzed (Fig. 6A; Table 1). Compared to wild type yeast, 42 proteins exhibited a statistically significant increase and 19 proteins demonstrated a significant decrease in abundance in *Ca_hrd1/Ca_hrd1* yeast. In *Ca_doa10/Ca_doa10* yeast, 39 proteins were significantly elevated, and 24 were significantly downregulated relative to wild type yeast. Consistent with a broader role for CaUbc7, a greater number of *C. albicans* proteins exhibited significant alterations in *Ca_ubc7/Ca_ubc7* yeast (82 increased, 63 decreased) compared to wild type yeast. The proteins with the largest significant increases in abundance in each mutant strain are presented in Table 2. 52% of proteins altered in *Ca_hrd1/Ca_hrd1* yeast, and 11% of proteins altered in *Ca_doa10/Ca_doa10* yeast, exhibited coordinated shifts in abundance in *Ca_ubc7/Ca_ubc7* yeast (Fig. 6B; Tables S3 and S4). Seven proteins exhibited significant, opposite-direction changes in abundance in multiple mutants (Table S5). Loss of *HRD1* caused a greater enrichment of predicted ER-targeted proteins (*i.e.*, proteins with signal peptides and/or transmembrane segments) than did loss of *DOA10* (Table 3).

A

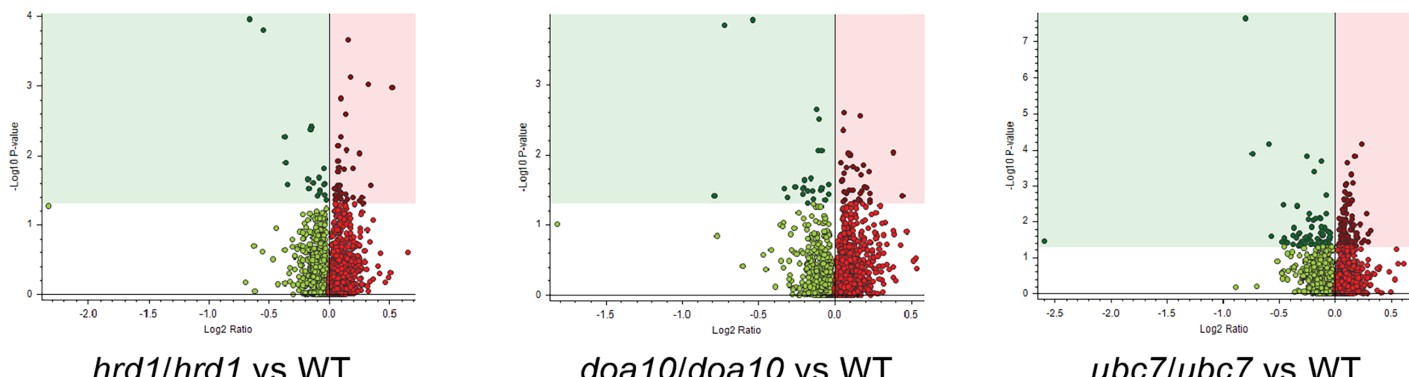

<p style="text-align:center">hrd1/hrd1 vs WT        doa10/doa10 vs WT        ubc7/ubc7 vs WT</p>

B

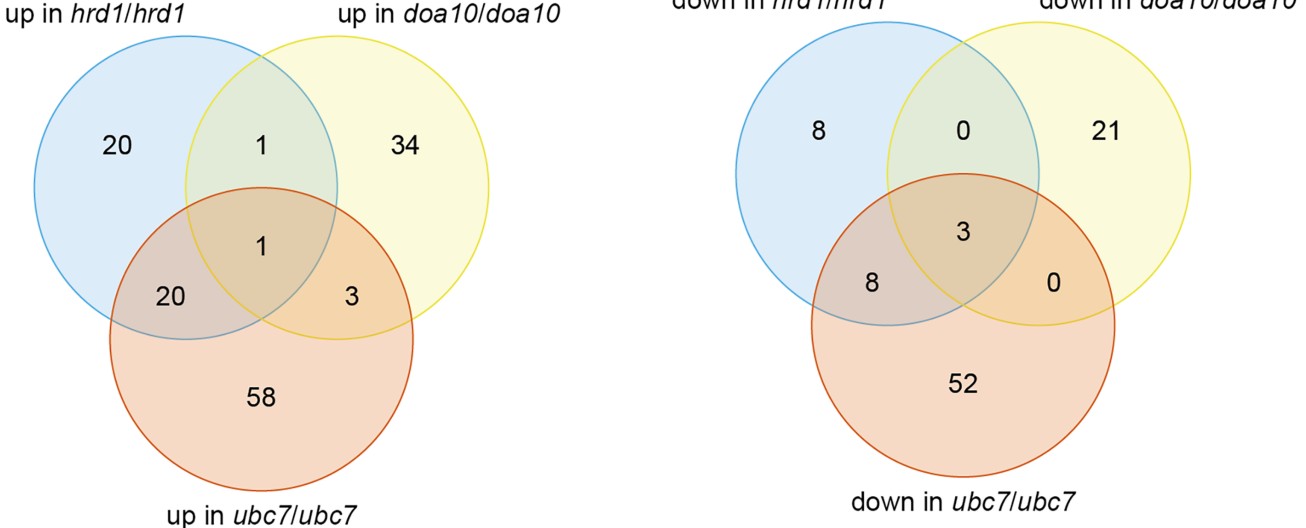

**Figure 6 Proteomic analysis of *C. albicans* ERAD mutants.** (A) Volcano plots illustrating changes in protein abundance between ERAD mutants and wild type *C. albicans vs* statistical significance of differences in abundance. Shaded regions in plots indicate data points for which $P < 0.05$. (B) Venn diagrams illustrating proteins present in increased (left) or decreased (right) abundance in indicated *C. albicans* mutants relative to wild type yeast.

**Table 1 Summary of proteins with altered abundance in mutant *C. albicans* relative to wild type.**

| | # of *C. albicans* proteins with significant differences in abundance | # of *C. albicans* proteins significantly increased in mutant | # of *C. albicans* proteins significantly decreased in mutant |
|---|---|---|---|
| *hrd1/hrd1 vs* WT | 61 | 42 | 19 |
| *doa10/doa10 vs* WT | 63 | 39 | 24 |
| *ubc7/ubc7 vs* WT | 145 | 82 | 63 |

**Table 2 Proteins with largest significant increases in abundance in each mutant strain.**

| hrd1/hrd1 | doa10/doa10 | ubc7/ubc7 |
|---|---|---|
| orf19.1186 | Rct1 (Ynl208w) | orf19.1186 |
| Rct1 (Ynl208w) | Ptk2 (Ptk2) | Sfp1 (Sfp1) |
| CR_06510W | Wal1 (Las17) | Sod3 (Sod2) |
| Tlo16 | Hsp70 (Ssa4) | orf19.5825.1 (Yos1) |
| Tlo34 | Cta26 | Tip20 (Tip20) |
| Tlo9 | Cta24 | orf19.252 (Mpc1) |
| Cta2 | Tlo1 | Rfc52 (Rfc5) |
| orf19.3140.1 (Ump1) | Tlo11 | Erg25 (Erg25) |
| Erg3 (Erg3) | Tlo8 | Oct1 (Oct1) |
| Taf145 (Taf1) | Tlo13 | Erg3 (Erg3) |

Note:
S. cerevisiae best hits or orthologs (per Candida Genome Database ortholog finder) are indicated in parentheses.

**Table 3 Predicted topology of proteins with altered abundance in C. albicans mutants.**

| | Increased abundance in hrd1/hrd1 | | Decreased abundance in hrd1/hrd1 | | Increased abundance in doa10/doa10 | | Decreased abundance in doa10/doa10 | | Increased abundance in ubc7/ubc7 | | Decreased abundance in ubc7/ubc7 | | Candida albicans proteome | |
|---|---|---|---|---|---|---|---|---|---|---|---|---|---|---|
| | # of proteins | % of proteins | # of proteins | % of proteins | # of proteins | % of proteins | # of proteins | % of proteins | # of proteins | % of proteins | # of proteins | % of proteins | # of proteins | % of proteins |
| TM | 6 | 14.3% | 1 | 5.3% | 3 | 7.7% | 2 | 8.3% | 13 | 15.9% | 15 | 23.8% | 973 | 15.6% |
| SP | 2 | 4.8% | 0 | 0.0% | 0 | 0.0% | 1 | 4.2% | 2 | 2.4% | 3 | 4.8% | 325 | 5.2% |
| TM and SP | 1 | 2.4% | 1 | 5.3% | 0 | 0.0% | 0 | 0.0% | 1 | 1.2% | 1 | 1.6% | 112 | 1.8% |
| Globular | 33 | 78.6% | 17 | 89.5% | 36 | 92.3% | 21 | 87.5% | 65 | 79.3% | 44 | 69.8% | 4,808 | 77.3% |
| Beta | 0 | 0.0% | 0 | 0.0% | 0 | 0.0% | 0 | 0.0% | 1 | 1.2% | 0 | 0.0% | 3 | 0.05% |
| Likely ER-targeted (TM + SP + TM and SP) | 9 | 21.4% | 2 | 10.5% | 3 | 7.7% | 3 | 12.5% | 16 | 19.5% | 19 | 30.2% | 1,410 | 22.7% |
| Total | 42 | 100.0% | 19 | 100.0% | 39 | 100.0% | 24 | 100.0% | 82 | 100.0% | 63 | 100.0% | 6,221 | 100.0% |

Note:
Proteins with altered abundance in C. albicans mutants were analyzed using the DeepTMHMM prediction tool (https://dtu.biolib.com/DeepTMHMM). TM, proteins with at least one predicted alpha-helical transmembrane segment. SP, proteins with predicted ER-targeting signal peptide. TM and SP, proteins with at least one predicted alpha-helical transmembrane segment and a predicted ER-targeting signal peptide. Globular, globular proteins without a predicted ER-targeting signal peptide or TM segment. Beta, protein with predicted transmembrane beta strands (i.e., beta barrel proteins).

We performed Gene Ontology (GO) analysis of genes encoding proteins with statistically significant changes in abundance in ERAD mutants (Tables S6–S8). Among proteins exhibiting increased abundance in Ca_hrd1/Ca_hrd1 yeast, genes with products that function in or are predicted to function in the ER and secretory pathway were enriched, including CaPdi1, CaKar2, CaDpm1, and CaSec61, homologs of proteins that are upregulated by the ER homeostatic unfolded protein response (UPR) in other organisms, including S. cerevisiae (Chapman, Sidrauski & Walter, 1998; Travers et al., 2000). GO analysis indicates a statistically significant enrichment of proteins mediating sterol synthesis in Ca_ubc7/Ca_ubc7 yeast (Table S8). The C-5 sterol desaturase ScErg3 is a bona fide ScHrd1/ScUbc7 substrate in S. cerevisiae (Jaenicke et al., 2011); enrichment in

both *Ca_hrd1/Ca_hrd1* and *Ca_ubc7/Ca_ubc7* mutants indicates CaErg3 is likely to be a physiological target of the *C. albicans* enzymes as well. Overall, our results suggest *C. albicans* ERAD pathway homologs play important roles in protein quality control and regulated protein degradation.

## DISCUSSION

To our knowledge, in this study, we conducted the first functional analysis of ERAD genes in a pathogenic fungus. We generated homozygous knockouts of *Ca_HRD1*, *Ca_DOA10*, and *Ca_UBC7* and compared stress resistance phenotypes and proteome composition among mutants and wild type yeast. Our results provide evidence for both quality control and regulatory function for *C. albicans* Hrd1, Doa10, and Ubc7.

Consistent with conserved roles in protein quality control, yeast with homozygous mutations in genes encoding CaHrd1, CaDoa10, or CaUbc7 exhibited sensitivity to hygromycin B and β-mercaptoethanol, which increase the burden of aberrant and misfolded proteins globally and in the ER, respectively. Enhanced sensitivity of mutants to proteotoxic stress at elevated temperatures (approximating human baseline and febrile temperatures) suggests a potential therapeutic vulnerability of *C. albicans*. It will be important to assess the impact of loss or inhibition of ERAD enzymes in animal models of infection in future studies.

Our proteomic analysis provides initial characterization of regulatory function of putative ERAD enzymes in *C. albicans*. Proteins with increased abundance in mutant yeast reflect candidate physiological ERAD substrates, ERAD co-factors, and other stress-response factors whose synthesis is induced as a compensatory response to defective ER protein quality control. We note additional biologically relevant proteins may exist that were not identified in this experiment; such proteins may be revealed by future analyses that include deeper offline fractionation for higher proteome coverage. Based on homology with other species, CaHrd1 and CaDoa10 are both expected to mediate ubiquitin transfer from CaUbc7 to substrate proteins. Consistent with shared substrates, we observed substantial overlap between proteins with altered abundance in *Ca_ubc7/Ca_ubc7* yeast and in *Ca_hrd1/Ca_hrd1* yeast and, to a lesser extent, between *Ca_ubc7/Ca_ubc7* yeast and *Ca_doa10/Ca_doa10* yeast (Fig. 6B; Tables S3 and S4).

In *S. cerevisiae*, a division of labor exists among ER protein quality control machinery such that Hrd1 promotes degradation of soluble ER luminal proteins and ER-targeted proteins that clog the Sec61 translocon, while Doa10 targets soluble cytosolic proteins. Both enzymes target misfolded or short-lived transmembrane proteins (*Mehrtash & Hochstrasser, 2019*). In our proteomic analysis, *HRD1* deletion caused greater enrichment of predicted ER-targeted proteins (*i.e.*, proteins with signal peptides and/or transmembrane segments) than did *DOA10* deletion (Table 3), consistent with a more heavily ER-biased clientele for CaHrd1 relative to CaDoa10.

Notably, predicted ERAD-linked proteins (*i.e.*, CaPdi1, CaKar2, CaDpm1, and CaSec61) were upregulated in both *Ca_hrd1/Ca_hrd1* and *Ca_ubc7/Ca_ubc7* yeast. ERAD disruption induces the UPR in other species, and homologs of these proteins are upregulated by the *S. cerevisiae* UPR (*Chapman, Sidrauski & Walter, 1998*; *Travers et al.,*

*2000*). Previous studies have shown that the UPR effector protein CaHac1 undergoes characteristic non-canonical splicing, regulates gene expression, and alters cell morphology in response to ER stress. Our results provide additional evidence for conservation of ER homeostatic mechanisms in *C. albicans* (*Wimalasena et al., 2008*).

Among proteins upregulated in putative ERAD-defective *C. albicans*, several possess homologs present in increased abundance in comparable *S. cerevisiae* proteomics experiments (*Foresti et al., 2014*). Four proteins upregulated in *Ca_hrd1/Ca_hrd1* cells have *S. cerevisiae* homologs exhibiting increased abundance in *Sc_hrd1* yeast (CaKar2, CaErg3, CaOrf19.1796/ScYpl113C, and CaOrf19.2346/ScFmp40). Likewise, three proteins upregulated in *Ca_ubc7/Ca_ubc7 C. albicans* have homologs exhibiting increased abundance in *Sc_ubc7* yeast (CaErg5, CaErg3, and CaErg25). One protein was upregulated in *Ca_doa10/Ca_doa10 C. albicans* for which an *S. cerevisiae* homolog was also found by mass spectrometry to be present at elevated levels in *Sc_doa10* yeast (CaVtc4).

The proteins with the greatest increases in abundance in *C. albicans* ERAD mutants do not possess obvious homologs in *H. sapiens*. For instance, the protein with the greatest increase in abundance in both *Ca_hrd1/Ca_hrd1* and *Ca_ubc7/Ca_ubc7* yeast was the product from the uncharacterized open reading frame, *Ca_orf19.1186* (*Ca_C6_00270W*). This protein has domains with predicted function in glycosylphosphatidyl inositol (GPI) anchor attachment to proteins, which occurs at the ER membrane. Orf19.1186 homologs are detectable in several *Candida* species, but not in *S. cerevisiae* or metazoans. Systematic analysis in S. *cerevisiae* has revealed both negative and positive genetic relationships between genes encoding ERAD machinery and those encoding GPI-biosynthetic enzymes (*Costanzo et al., 2019*; *Costanzo et al., 2016*). The mechanisms underlying these interactions are unknown; it has been hypothesized that misfolded proteins that accumulate in ERAD mutants impact the GPI pathway (*Nakatsukasa, 2021*).

The protein with the greatest increase in abundance in *Ca_doa10/Ca_doa10* yeast (and the second greatest increase in abundance in *Ca_hrd1/Ca_hrd1* yeast) is encoded by *Ca_RCT1*, an uncharacterized ORF. The molecular function of CaRct1 is unknown, but the poorly characterized *S. cerevisiae* homolog (ScYnl208w) has been found to associate with ribosomes (*Fleischer et al., 2006*). *Ca_RCT1* homologs are detectable in multiple fungal, archaeal, and bacterial species, but not in metazoans. CaOrf19.1186 and CaRct1 may be ERAD substrates in *C. albicans*. Alternatively, increased abundance of these proteins may reflect enhanced synthesis of compensatory response factors due to loss of ERAD enzymes.

In *S. cerevisiae* and humans, ERAD machinery controls sterol abundance to meet cellular demands *via* feedback-regulated degradation of sterol biosynthetic enzymes (*Foresti et al., 2013*; *Garza, Tran & Hampton, 2009*; *Hampton, Gardner & Rine, 1996*; *Huang & Chen, 2023*; *Jo et al., 2011*; *Stevenson, Huang & Olzmann, 2016*). We observed enrichment of proteins with roles in sterol synthesis in *C. albicans* ERAD mutants, including CaErg3, the homolog of a *bona fide S. cerevisiae* ERAD substrate (*Jaenicke et al., 2011*). These results suggest sterol synthesis is also regulated by ERAD in *C. albicans*. Two distinct aspects of ergosterol biology are targeted by antifungal therapeutic agents, including amphotericin B and azoles. This link could provide avenues for development of

novel antifungal drugs and identification of novel drug targets. Our findings highlight the need for future studies to validate and characterize the relationship between *C. albicans* ERAD biology and sterol production and to interrogate molecular similarities and differences of ERAD mechanisms in humans and *C. albicans*.

## CONCLUSIONS

To our knowledge, we provide the first-ever characterization of genes predicted to encode ERAD enzymes in a pathogenic fungus. Homozygous deletion of *Ca_HRD1*, *Ca_DOA10*, or *Ca_UBC7* sensitized *C. albicans* to proteotoxic stress and altered the proteome in distinct, but overlapping, ways. Among other perturbations, ERAD disruption increased the abundance of sterol-biosynthetic enzymes. These results strongly suggest ERAD machinery performs both protein quality control and regulatory functions. Future work will be conducted to characterize putative physiological substrates, co-factors, and compensatory stress response factors that exhibited altered abundance in *C. albicans* ERAD mutants. Heightened sensitivity of *C. albicans* ERAD mutants to proteotoxic stress at physiologically relevant temperatures suggests a potential therapeutic vulnerability. The role of ER homeostatic mechanisms in moderating virulence will be investigated in subsequent studies.

## ACKNOWLEDGEMENTS

The mass spectrometry performed in this work was completed by the Indiana University School of Medicine (IUSM) Proteomics Core. We thank Guihong Qi and Amber Mosley for outstanding advice and technical assistance.

### Funding

This work was funded by National Institutes of Health R15 grants GM111713 (Eric M Rubenstein) and AI130950-02 (Douglas A Bernstein), an Indiana Academy of Science Senior Research grant (Douglas A Bernstein), and a Ball State University Graduate Student Aspire Research grant (Ellen M Doss). Acquisition of the IUSM Proteomics Core instrumentation used for this project was provided by the Indiana University Precision Health Initiative. The proteomics work was supported by the Indiana Clinical and Translational Sciences Institute, National Institutes of Health (Award Number UL1TR002529), a National Center for Advancing Translational Sciences, Clinical and Translational Sciences Award, and the Cancer Center Support Grant for the Indiana University Simon Comprehensive Cancer Center (Award Number P30CA082709) from the National Cancer Institute. The funders had no role in study design, data collection and analysis, decision to publish, or preparation of the manuscript.

### Grant Disclosures

The following grant information was disclosed by the authors:
National Institutes of Health R15 grants GM111713 and AI130950-02.

Indiana Academy of Science Senior Research.
Ball State University Graduate Student Aspire Research.
Indiana University Precision Health Initiative.
Indiana Clinical and Translational Sciences Institute.
National Institutes of Health: UL1TR002529.
National Center for Advancing Translational Sciences.
Clinical and Translational Sciences Award.
Indiana University Simon Comprehensive Cancer Center: P30CA082709.
National Cancer Institute.

## Competing Interests

Ellen M. Doss is employed by the Mode of Action and Resistance Management Center of Expertise at Corteva Agriscience.

## Author Contributions

- Ellen M. Doss performed the experiments, analyzed the data, authored or reviewed drafts of the article, and approved the final draft.
- Joshua M. Moore performed the experiments, analyzed the data, authored or reviewed drafts of the article, and approved the final draft.
- Bryce H. Harman performed the experiments, analyzed the data, authored or reviewed drafts of the article, and approved the final draft.
- Emma H. Doud performed the experiments, analyzed the data, prepared figures and/or tables, authored or reviewed drafts of the article, and approved the final draft.
- Eric M. Rubenstein conceived and designed the experiments, analyzed the data, prepared figures and/or tables, authored or reviewed drafts of the article, and approved the final draft.
- Douglas A. Bernstein conceived and designed the experiments, performed the experiments, analyzed the data, prepared figures and/or tables, authored or reviewed drafts of the article, and approved the final draft.

## Data Availability

The data are available at MassIVE: MSV000091843, DOI 10.25345/C58P5VK93.

## Supplemental Information

Supplemental information for this article can be found online at http://dx.doi.org/10.7717/peerj.15897#supplemental-information.

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
