# Peer review of "Characterization of endoplasmic reticulum-associated degradation in the human fungal pathogen Candida albicans"

_PeerJ, doi:10.7717/peerj.15897_

## Round 0.1 · original submission · Minor Revisions

Dear Dr. Bernstein

I am pleased to inform you that your manuscript has been well received by the reviewers and only requires minor revisions. Upon resubmission of your revised manuscript,, please be sure to specify your response to each of the reviewer's comments. Thank you for submitting your work to PeerJ and we look forward to receiving your revised manuscript.

Reviewer 1 ·

Basic reporting

This manuscript offers a comprehensive examination of the functional role of ERAD genes in C. albicans, a significant pathogenic fungus. The scientific language employed is appropriate for the target audience, and the contextualization of this research within the broader field is commendable. The study's conclusions are clearly articulated and provide a suitable springboard for future work.

However, the presentation of results, while detailed, could benefit from a more streamlined approach. The authors are advised to reconsider the necessity of each figure and table. It might be advantageous to combine some or relegate less central ones to the supplementary material. This adjustment could enhance the narrative's flow and improve readability without compromising the data's integrity.

Experimental design

The researchers demonstrate thoughtful planning and rigorous methodology in their experimental design. The use of homozygous gene knockouts and proteotoxic stress inducers aligns well with the study's aims. Moreover, the successful execution of TMTpro label-based proteomic profiling provides essential insights into ERAD enzymes' function and potential substrates.

However, the protein identification process needs further clarification. The supplemental data reveals that 755 out of 3479 identified proteins were discovered with less than two unique peptides, which is less than ideal for accurate protein identification. The authors should address this issue to guarantee the results' validity and the interpretations that follow.

Validity of the findings

The study's results are substantial and well-supported by the presented data. The proteomic analysis has provided significant insights into the potential substrates of the ERAD enzymes and affirmed the dual roles these proteins play in protein quality control and regulation.

The discovery of heightened sensitivity in C. albicans ERAD mutants to proteotoxic stress under physiological temperatures suggests potential therapeutic targets. This finding reinforces the clinical relevance of this study. However, inconsistencies in the proteomic data could undermine these compelling findings. The issue of 233 proteins identified but devoid of abundance values for any TMTpro channels must be addressed to validate the data and strengthen the study's conclusions.

Additional comments

The authors are urged to address specific caveats in the discussion, including potential errors in protein isolation, label-based quantification at the peptide level, or mass spectrometry. The analysis and interpretation of the data lean heavily on the assumption that increased protein abundance is due to reduced degradation, neglecting the possible role of enhanced protein synthesis as a compensatory mechanism. Furthermore, the relationship between ERAD biology and sterol biosynthesis, while intriguing, requires further validation.

The physiological significance of the ERAD mutants' sensitivity to proteotoxic stress at elevated temperatures should be examined in more detail, perhaps using animal models of C. albicans infection. Similarly, the significance of protein homologs in S. cerevisiae and their absence in H. sapiens needs further exploration.

In conclusion, this study is a significant step toward understanding the functional role of ERAD proteins in pathogenic fungi. It sets the stage for future investigations that could translate these findings into novel therapeutic strategies for fungal infections. Addressing the identified limitations will further strengthen the study and its potential implications.

Reviewer 2 ·

Basic reporting

Doss et al generated three knockouts of fungi Candida albicans genes encoding Hrd1 Doa10 and Ubc 7 that were predicted to have essential function ERAD. The fitness of the mutant under proteotoxic or ER redox stress conditions demonstrates the influence of each encoding gene. Furthermore, each mutant's proteomics analysis provides insights into its functions in the ERAD process. In general, the manuscript is concise and clear, with no major concerns with reference, background, experiment design, or data interpretation. There are some questions or suggestions in the following section.

Experimental design

Please provide references for inhibition analysis and include the experimental conditions applied in the manuscript.

Validity of the findings

Figure 4. Line 278-279: " Heterozygous Ca_HRD1/Ca_hrd1 yeast exhibited wild type
resistance to hygromycin B, indicating a single copy of Ca_HRD1 is sufficient to confer protection from hygromycin B." Under 40C, heterozygous Ca_HRD1/Ca_hrd1 has lower resistance to hygromycin B than homozygous, therefore the single copy of Ca_HRD1 is less sufficient under higher temperatures.

Reviewer 3 ·

Basic reporting

Authors generated three Candidate albicans mutations with homozygous Hrd1, Doa10, and Ubc7 gene deletions, which are involved in ER-associated protein degradation, in this the manuscript entitled “Characterization of endoplasmic reticulum-associated degradation in the human fungal pathogen Candida albicans”. Authors found that these mutants become hypersensitive to proteotoxic stress and temperatures and well described the procedure of the experiments. The study is interesting.

Minor concerns:

In Figure 6A and 6B, authors may list the major altered targets induced by the mutants, which may help readers to follow the findings in this study.

Authors may check the typing in the figure legends including Figure 6A: “f” in “significance”; Figure 5: sterile xlter.

Experimental design

well described

Validity of the findings

interesting

---

## Round 0.2 · accepted · Accept

I am happy to inform you that your manuscript has now been accepted for publication in PeerJ. Thank you for addressing the comments of the reviewers and we are happy that you selected PeerJ for publication. We look forward to future manuscripts from you for consideration by the journal.

Reviewer 1 ·

Basic reporting

No more comments.

Experimental design

No more comments.

Validity of the findings

No more comments.

Additional comments

The authors have effectively responded to all the points raised and have suitably revised the manuscript. In my opinion, the manuscript is now in a state suitable for publication.

Reviewer 2 ·

Basic reporting

The manuscript has been changed according to the reviewer's suggestions and the experiment proposed by the reviewer requires to be conducted in different animal models, thus authors included reviewers' suggestions in the discussion section. Recommend accepting the paper.

Experimental design

n/a

Validity of the findings

n/a

Reviewer 3 ·

Basic reporting

Authors addressed my questions.

Experimental design

sufficient

Validity of the findings

meaningful